# Optimized Downlink Scheduling over LTE Network Based on Artificial Neural Network

**Falah Y. H. Ahmed** [1,*], **Amal Abulgasim Masli** [2], **Bashar Khassawneh** [3], **Jabar H. Yousif** [1] **and Dilovan Asaad Zebari** [4]

[1]  Faculty of Computing and Information Technology, Sohar University, Sohar 311, Oman; jyousif@su.edu.om
[2]  Faculty of Education, Computer Science Department, Misurata University, Misrata 9329+V25, Libya; amalabulgasim@gmail.com
[3]  Department of Computer Science, Irbid National University, Irbid 2600, Jordan; b.khassawneh@inu.edu.jo
[4]  Department of Computer Science, College of Science, Nawroz University, Duhok 42001, Iraq; dilovan.majeed@nawroz.edu.krd
[*]  Correspondence: fhamode@su.edu.om

**Abstract:** Long-Term Evolution (LTE) technology is utilized efficiently for wireless broadband communication for mobile devices. It provides flexible bandwidth and frequency with high speed and peak data rates. Optimizing resource allocation is vital for improving the performance of the Long-Term Evolution (LTE) system and meeting the user's quality of service (QoS) needs. The resource distribution in video streaming affects the LTE network performance, reducing network fairness and causing increased delay and lower data throughput. This study proposes a novel approach utilizing an artificial neural network (ANN) based on normalized radial basis function NN (RBFNN) and generalized regression NN (GRNN) techniques. The 3rd Generation Partnership Project (3GPP) is proposed to derive accurate and reliable data output using the LTE downlink scheduling algorithms. The performance of the proposed methods is compared based on their packet loss rate, throughput, delay, spectrum efficiency, and fairness factors. The results of the proposed algorithm significantly improve the efficiency of real-time streaming compared to the LTE-DL algorithms. These improvements are also shown in the form of lower computational complexity.

**Keywords:** LTE network; resource allocation; ANN; normalized model

## 1. Introduction

The 3rd Generation Partnership Project (3GPP) implemented the (LTE) strategy for fulfilling the increasing demand for wireless networks. The radio resource management scheduling algorithms were also used in this LTE setup. They assigned radio services to the final users based on different criteria for quality of service (QoS). However, the development of the downlink scheduling algorithms was a major problem noted during resource allocation in the LTE system. Different scheduling techniques were proposed for addressing this issue, and an investigation of the downlink algorithms garnered a lot of research interest during the LTE implementation, as several researchers started shifting to packet scheduling over LTE since it was regarded as a rapidly growing technology that can significantly affect the future of wireless networks. In the LTE downlink algorithm used for QoS class identifiers, the radio resource allocation steps use the QoS specifications and channel condition reports to determine the users' transmission orders. However, inefficient resource allocation in the LTE networks can be noted due to poor network performance that deteriorates the data throughput and network fairness index and increases the average delay.

A better communication system could be developed by integrating the different artificial intelligence (AI) concepts and machine learning (ML) techniques for scheduling the end-user devices and designing wireless structures. The use of artificial intelligence

technologies in the wireless communication system was based on its ability to encourage the massive advancement of wireless traffic and the emergence of new uses for wireless services. The services that were not tested in the past varied between general multimedia and video-based services. The major issue in the evolution and development of wireless networks in the past decade is a greater need for wireless networks that provide better communication services with higher reliability, lower latency, and a higher end rate [1].

This study focuses on resource allocation scheduling at the receiver end for optimizing the downlink across the LTE network. It ensures the fulfilment of the QoS about their fairness. For this purpose, the study investigated three artificial neural network (ANN) algorithms, i.e., the approximate radial basis function neural network (RBFNN), the exact RBFNN, and the generalized regression neural network (GRNN). The researchers approximated the exact Gaussian fields using the appropriate radial function (RF). ANNs perform many tasks, like approximation or function prediction, pattern classification, and clustering. The performance of the ANN during tasks is significantly affected by the data preparation setup of some of the ANN structures [2,3]. In previous studies, the researchers investigated the performance of the ANN concerning wireless communication [4]. They noted that ANNs could be used for studying network prediction and user behavior and offering solutions to users for different problems associated with wireless networks. These problems included resource computation, spectrum management, replacing cached contents, cellular connectivity, and resource assignment.

The remaining paper has been organized in the following manner: Section 2 presents an overview of related studies that used different ANN models for the LTE network. Section 3 presents a review of the LTE downlink scheduling techniques. Section 4 describes the architecture of the proposed ANN model that processes the LTE downlink scheduling algorithms. It then follows with a description of the simulation environment and results. The final section presents the conclusions.

## 2. Literature Review

Many researchers have used ANN models to enhance the performance of the downlink communication system in LTE networks. Different models were used, where some studies considered channel estimation, while others investigated user device condition or mobile location. Predictive analyses of various ANN models for predicting and classifying the data over the LTE network were conducted in some other studies. Several studies have reported significant improvements in various performance metrics, such as throughput, fairness, and quality of service, when using ANNs for downlink scheduling. For example, a study by [5] showed that an ANN-based downlink scheduling algorithm achieved up to 30% higher throughput than a conventional rule-based algorithm.

Charrada [6] proposed an accurate channel environment estimation technique using the ANN and support vector machine regression (SVR) models for the standardized signal structure of the LTE downlink system [6]. This technique was used under the impulsive, non-linear noise that interfered with reference codes after considering the high mobility conditions. He studied the SVR and ANN performances using the simulation results, which performed better than the decision feedback (DF), least squares (LS), and ANN algorithms. Another study [7] presented a method for relaying the reference symbol information. This information was used for estimating the total frequency response of the channel. This technique was summarized in two steps: firstly, channel differences were adapted after applying the ANN-based learning methods trained using the genetic algorithm (ANN-GA). Secondly, the channel matrix was estimated to improve the performance of LTE networks. They validated the proposed algorithms using various ANN-based estimator algorithms, such as the feed-forward neural network, the layered recurrent neural network, the least squares (LS) algorithm, and the cascade-forward neural network for closed-loop spatial multiplexing (CLSM) single-user multi-input. The results of this comparison indicated that the proposed ANN-GA algorithm showed better accuracy than others.

Furthermore, the significant increase in network subscribers led to resource allocation issues. To resolve this problem, the researcher proposed a downlink algorithm that ensured an effective and faster resource allocation solution for real-time video applications [8]. This solution used an ANN algorithm that allowed resource allocation after considering the UE conditions. They noted that the AI techniques used for resource allocation on the LTE network generated accurate results, but the ANN-based training process could take a long time. Hence, dynamic resource allocation can be done by realizing the daily ANN training processes whenever the eNodeB is intense.

In terms of predicting and classifying data over the LTE network via applying ANN models, many researchers have carried this out. In an earlier study [9], the researchers investigated the performance of two ANN models for prediction and training algorithms (i.e., Levenberg-Marquardt and Bayesian regularization). They primarily focused on integrating an ANN into the LTE network during the mobile handover start-up phase. They compared the received signal strength (RSS) and the hysteresis fringe parameters for the adaptive neural hysteresis fringe reduction algorithm. The study aimed to resolve the channel estimation problem noted in LTE networks. In [10], the researchers determined the adaptive learning and predictive ability of three ANN models, i.e., RBFNN, GRNN, and MLPNN, using the spatial radio signal dataset derived from the commercial LTE cellular networks. Thus, they could verify the efficiency and accuracy of the adaptive prediction system using attenuation and oscillation landscapes to determine the radio signal strength propagated in the LTE urban micro-cell topography. Their results indicated that the ANN prediction techniques could adapt to the measurement errors regarding the attenuation of the LTE radio signals. A comparison of the performance of the different techniques indicated that all ANN models could predict the transmitted LTE radio signals with numerous errors. A recent study by Ojo [11] aimed to resolve these issues related to the existing models (experimental and deterministic) by implementing ML-based algorithms to predict path loss in LTE networks. They developed the RBFNN and multilayer perception neural network (MLPNN) models with the measured data as an input variable and compared it to the measured path loss. They noted that the RBFNN was more accurate, as it showed lower root mean squared errors (RMSEs) than the MLPNN. Also, an ANN-aided scheduling scheme for the UEs with mobility in LTE dynamic HetNets was proposed in another study [12]. They determined a faster eICIC reconfiguration technique in the LTE HetNets. This technique helped to achieve a marginal gap compared to the centralized solution. Using historical data, this proposed technique could train the RBFNN to determine the relationship between the surrounding environment (channel and UE deployment), an optimal cell range extension, and a nearly blank subframe pattern. The researchers investigated the performance of their proposed algorithm concerning its throughput and utility during simulations. They noted that an optimal resource assignment helped rapidly vary the HetNets with low-performance degradation. In [13], the researchers studied the probabilistic GRNN model for modelling and estimating the data corresponding to the spatial signal power loss. Commercial data was collected from the LTE network interface's outdoor location. This examined GRNN model was trained with a power loss measurement of a spatial signal. The data was collected from three different outdoor signal propagation locations. It is noted that the proposed model showed better results in comparison to the conventional least square regression modelling process. Some researchers proposed an ANN-based classification system with higher accuracy and performance [14] for fingerprint images using the NN models. In [15], the researchers used this technique for classifying bacteria. The results indicated that the ANN was effective and feasible. In [16], the researchers classified aerial photographs using the ANN. They noted that the ANN was suitable for classifying the remotely sensed data, exceeded the maximal probability classification for the classification accuracy, and showed a positive effect. In [17], the researchers used the ANN for classifying spoken letters. They noted a 100% (training) and 93% (testing) classification accuracy. All these studies stated that the ANN could be used for classification owing to its better performance. To improve machine-type communication (MTC) security, some

researchers [18] studied the issues related to the lack of authentication requirements and attack detection for LTE-based MTC devices. As a result, they introduced a better NN model for detecting attacks. The results indicated the efficiency of this model in detecting attacks and compared the system's security.

ANN models could be used for optimizing the resource allocation algorithms for communication networks. Furthermore, ANN technology could overcome the resource allocation problems noted in the LTE network. Some common ANN activation functions that were applied included the sigmoidal, binary, and hyperbolic sigmoidal functions based on the RBFNN, MLPNN, recurrent neural network, and perceptron models. All these functions were used for developing network communication. The researchers also considered the backpropagation and the gradient descent algorithms as training algorithms for the ANN. ANNs are used for different tasks, like approximation and prediction of functions, pattern classification, prediction, and clustering. However, the performance of the algorithm was significantly affected by data preparation and the setup used for the NN structure. ANN and mathematical models are used to evaluate and validate experimental data [19,20]. Artificial neural network (ANN) techniques are used in the GRNN-RBFNN model and other proposed methods to optimize downlink scheduling over LTE networks. These models use historical data to learn and predict optimal scheduling policies for different users. However, some challenges still need to be addressed in LTE network scheduling. One such problem is ensuring fairness among users with different QoS requirements. The scheduling algorithms used in these models may not always be able to provide equal QoS to all users, leading to a lower fairness index for some users. Therefore, further research is needed to develop scheduling algorithms to ensure fairness among users with varying QoS requirements. Another challenge is the dynamic nature of the LTE network, which may lead to fluctuations in network conditions and user demands. This can make it difficult to predict optimal scheduling policies accurately. Therefore, there is a need to develop adaptive scheduling algorithms that can adjust to changes in various network channels' conditions and user demands in real time considering the data over the network. The proposed methods have shown promise for optimizing downlink scheduling on LTE networks. To improve the performance of LTE network scheduling, the problems of fairness and changing network conditions still need to be studied.

The proposed approach of using artificial neural networks (ANNs) for downlink scheduling in LTE networks is suitable for solving the critical problem of optimizing the allocation of radio resources to users. ANNs are powerful machine-learning models that can learn from historical data to identify patterns and relationships. In the last few years, many researchers have used ANN models to enhance the performance of the downlink communication system in LTE networks. Many models have been used; some studies considered channel estimation, while others investigated the condition of the user devices or estimated the mobile location. Predictive analyses of various ANN models for predicting and classifying the data over the LTE network were conducted in some other studies. Several studies have reported significant improvements in various performance metrics, such as throughput, fairness, and quality of service, when using ANNs for downlink scheduling.

## 3. Downlink Scheduling in LTE

Dynamic resource allocation, called the packet scheduler algorithm, is an essential feature in network communication and aids in controlling the selection of RBNN for the user equipment (UE) to prevent signal interference. A scheduler algorithm helps to acquire an adequate allocation of physical resource blocks (PRBs) (e.g., frequency, power, time, etc.) to the UEs that fulfil the QoS requirements, according to scheduling standards, for enabling a fair distribution of the available resources amongst the users [21]. The downlink scheduling algorithms are described using the metric $M_{i,k}$, in which resources were allocated to each UE based on the parallel between the metrics $i$ of the users who had the highest $M_{i,k}$ values and the $k$ RBFNN that was allocated. The popular downlink

scheduling algorithms have been described below and categorized based on their service for real-time and non-real-time applications.

### 3.1. Proportional Fair (PF) Algorithm

The PF technique allocates the users' existing available radio resources. It considers the general channel characteristics and previous data concerning throughput as the weighting factor for the predicted data rate. The objective of the PF technique is to maximize the overall throughput while also providing fair data flow, as in Equation (1) [22].

$$M_{i,k}^{PF} = \frac{d_{i,k}(t)}{R_i(t)} \tag{1}$$

This measure ascertains the fraction of the presently accessible data rate $d_{i,k}(t)$, and the average of the previous data rate $R_i(t)$, while $i$ corresponds to the flow in the $k$ sub-channel.

### 3.2. Maximum Largest Weighted Delay First (MLWDF) Metric

This channel-aware technique serves several data access streams with different QoS needs. MLWDF considers fair distribution, delay reduction, and ensuring system throughput. This algorithm handles real-time and non-real-time data flow, wherein the PF is used for the non-real-time flow, and the following weighting expression is used for the real-time data flow by applying the following weighting metrics, as in Equation (2).

$$M_{i,k}^{M-LDWF} = \propto_i \; D_{HOL,i} * M_{i,k}^{PF} \tag{2}$$

$$\propto_i = -\frac{log\delta_i}{\tau_i}$$

wherein, $D_{HOL,\,i}$ denotes the head-of-line delay of packet faced at time t, for the $i$ user, $\tau_i$ denotes the delay threshold for a packet for each user $i$ that was considered for real-time data, and $\delta_i$ denotes the maximal possibility of an HOL packet delay that may cause the user $i$th to exceed their threshold delay [23].

### 3.3. Exponential/Proportional Fairness (EXP/PF)

The EXP/PF technique has been formulated to support multimedia services for systems using multiplexed time. It can reduce package transmission times by combining the PF algorithm's features and the exponential function. The PF algorithm [22] is used to promote the infinite buffer flow, whereas the EXP/PF can be used to serve real-time traffic, as in Equation (3).

$$M_{I,K}^{EXP/PF} = \exp\left(\frac{\alpha_i D_{HOL,i} - x}{1 + \sqrt{x}}\right) \cdot \frac{d_k^i(t)}{R^i(t-1)} \tag{3}$$

where

$$x = \frac{1}{N_{rt}} \sum_{i=1}^{N_{rt}} \alpha_i D_{HOL,i}$$

### 3.4. Frame Level Scheduler (FLS)

This scheduler uses the two-level structure to ensure a specified delay value interval between the real-time data flow. The transmission of the total data in the actual time flow is determined at the highest level of the scheduler structure. The lower level works on every TTI and assigns RBs per flow. In this algorithm, the bandwidth requirements were estimated at the upper level. Then, the PF method was employed for sharing the reserve spectrum amongst the users showing the best efforts [24]. The researchers calculated the amount of data transmitted using Equation (4).

$$u_i\,(k) = h_i(k) * q_i(k) \tag{4}$$

where $u_i(k)$ refers to the quantum of data moved to correspond to the *i* flow and the *k* frame. This was acquired by transmitting a $q_i(k)$ queue level signal over a linear time-invariant filter with response $h_i(k)$; denoting discrete-time convolution [22].

## 4. Proposed Model

In this study, the researchers have focused on normalizing the data for the LTE downlink scheduling algorithms, which could improve their resource allocation performance concerning fairness. This was performed with the assistance of the proposed ANN model. This model included some initial phases, including:

i. Determining the data in the LTE downlink system that required treatment using the ANN models;
ii. Pre-processing processes used for excluding the conflicting data;
iii. Data partitioning, wherein the data was separated into two parts (i.e., training and testing);
iv. Assigning these data a number of input-output nodes, hidden nodes, and the transfer function for the topology of the ANN model, which explains the association between nodes and the choice of the ANN architecture;
v. Network training that considers experimentation to validate the proposed network model using the proposed algorithm and data classification; and
vi. Finally, the model selection, which includes a selection technique based on overall performance.

The technique's primary goal was to determine the diagnostic accuracy network model. Figure 1 demonstrates the framework of the proposed ANN normalized model used for the LTE downlink scheduling algorithm.

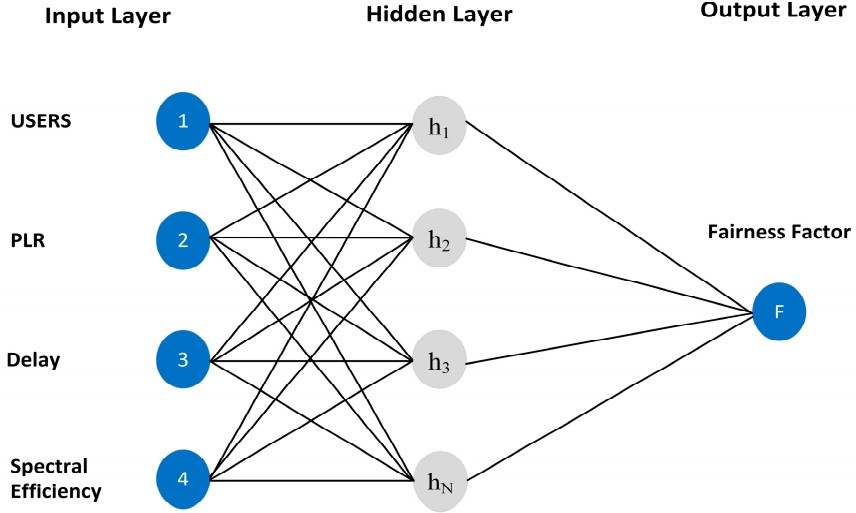

**Figure 1.** Framework for the ANN Topology Concept.

### 4.1. The Framework of the Proposed Model

This model includes four input nodes: users, packet loss rate (PLR), delay, and spectral efficiency, along with one output node (i.e., fairness factor). Moreover, selecting samples for the training and testing sets is critical to ensuring the ANN model's accuracy and generalization capability. A common approach is to randomly split the data into a training set and a testing set, with a typical ratio of 80% for the training set and 20% for the testing set. However, other ratios can also be used based on the specific problem's requirements in techniques such as cross-validation to ensure that the ANN model's performance is consistent across different datasets. In order to ensure that the ANN model's results adequately reflect its ability to work with collected data, it is essential to use appropriate training and testing techniques. Overfitting occurs when the ANN model is too complex and fits the

training data too closely, leading to poor generalizations on new data. Techniques such as early stopping, regularization, or dropout can be used to avoid overfitting.

Furthermore, the performance of the ANN model in this paper is evaluated on a separate validation dataset to ensure that it can generalize well to new data. The number of hidden nodes in the ANN model determines its complexity, which can significantly impact its performance. A model with too few hidden nodes may not capture the underlying patterns in the data, while a model with too many hidden nodes may overfit the training data. Therefore, it is essential to find the optimal number of hidden nodes that balances the model's complexity and its ability to capture the underlying patterns in the data, as shown in Figure 1.

The mean square error (MSE) determines the ANN model's training performance. It helps to decrease the number of mathematical operations and the memory needed for any computer task. Furthermore, it decreases the MSE value of the training data. MSE states that the neurons in the hidden layer are similar to the number of rounds required during the training process while executing the ANN model [25], as the MSE described in Equation (5).

$$MSE = \frac{1}{2} \sum_{i=1}^{R} (t(n) - a(n))^2 \tag{5}$$

where $(a(n))$ refers to the actual output generated by a network, $(t(n))$ indicates the expected output, and R refers to the number of rounds. This study presented a scheduling algorithm dataset with 50 data samples. The proposed ANN model is described above. Figure 2 indicates the framework steps for the proposed normalizing data based on the ANN model.

The method used for normalizing input data for the ANN model is designed, which is how the data that goes into it is normalized. However, the methods for normalizing data include the following:

Min-max scaling: This method scales the data to a fixed range, typically between 0 and 1. The formula used for this scaling is Equation (6).

$$x\_scaled = (x - min(x)) / (max(x) - min(x)) \tag{6}$$

Standardization: This method scales the data to have zero mean and unit variance. The formula used for this scaling is Equation (7).

$$x\_scaled = (x - mean(x)) / std(x) \tag{7}$$

Max normalization: This method scales the data such that the maximum value is 1. The formula used for this scaling is Equation (8).

$$x\_scaled = x / max(x) \tag{8}$$

The choice of normalization method depends on the data distribution and the specific problem. In some cases, it may be beneficial to use a combination of these methods, such as applying standardization after using min-max scaling. Normalizing the input data for ANN models is crucial to ensure that the features are on the same scale, which helps prevent the model from giving undue importance to features with higher values. Additionally, normalization can speed up the training process by reducing the oscillations in the loss function during training.

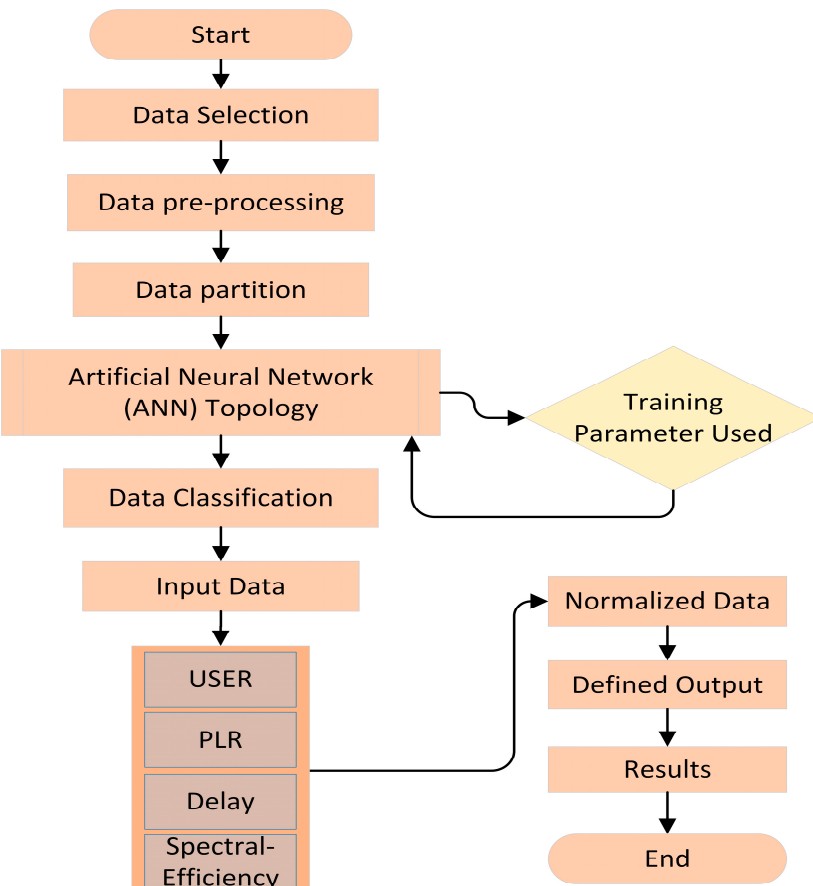

**Figure 2.** Proposed ANN Processing Framework Technique for Normalizing the Data.

*4.2. Proposed ANN Models for LTE Downlink Scheduling*

Solving these issues related to the allocations improves the video flow over the LTE network. We have described three ANN models used for normalizing the LTE system data. These included the approximate RBFNN (ARBFNN), the exact RBFNN (ERBFNN), and the generalized regression neural network (GRNN). There are three layers within the networks under the radial basis function neural network (RBFNN), including an input layer, a non-linear hidden layer with an activation function of RBF, and the linear output layer [26]. Furthermore, input modelling can be achieved as an absolute number vector $(x \in \mathbb{R}^n)$. In contrast, the network output represents a scalar function from the input vector, $(\varphi : \mathbb{R}^n \to \mathbb{R})$ which is presented as Equation (9).

$$\varphi(x) = \sum_{i=1}^{N} a_i \rho(\| x - c_i \|) \tag{9}$$

where $(N)$ refers to the hidden layer neuron number, $(c_i)$ denotes the neuron's center vector, $(i)$, and $(a_i)$ indicates the neuron's weight $(i)$ used in the neuron's output linearly. Consequently, functions dependent on the center vector distance are symmetrically radial towards such a vector, resulting in the nomenclature "radial basis functions" [26]. Regarding the basic form, individual hidden neurons are linked to the inputs directly. The RBF is regarded in its Gaussian as Equation (10).

$$\rho(\| x - c_i \|) = exp\left[ -\beta \| x - c_i \|^2 \right] \tag{10}$$

These Gaussian basis functions are seen to be local to their center vectors as follows: $\lim_{||x|| \to \infty} \rho(\| x - c\_i \|) = 0$. The RBFNN can be normalized, and the mapping is shown as Equations (11) and (12).

$$\varphi(x) \underline{\underline{def}} \frac{\sum_{i=1}^{N} a_i \rho(\| x - c_i \|)}{\sum_{i=1}^{N} \rho(\| x - c_i \|)} = \sum_{i=1}^{N} a_i u(\| x - c_i \|) \tag{11}$$

where

$$u(\| x - c_i \|) \underline{\underline{def}} \frac{\rho(\| x - c_i \|)}{\sum_{i=1}^{N} \rho(\| x - c_i \|)} \tag{12}$$

All the ANNs are trained for estimating the posterior probabilities of class membership by mixing the hyperplanes and the Gaussian basis functions [27]. This is called the "normalized radial basis function," whereas the "RBFNN" was utilized to classify data. Moreover, the generalized regression NN (GRNN) model was seen to be an interpretation of the RBNN model [28]. GRNN could include purposes of prediction, classification, and regression of target data. Thus, it could serve as a reliable solution for dynamic online systems. It depicts a better NN technique that is based on nonparametric regression. The model is based on the idea that each training sample acts as a mean for the radial basis neurons [28]. Mathematically, the GRNN is represented as Equation (13).

$$y(x) = \frac{\rho(\| x - c_i \|)}{\sum_{k=1}^{N} k(x, x_k)} \tag{13}$$

where $Y(x)$ denotes the value predicted for the $x$ input; K refers to the weight activation of a pattern layer neuron at $k$; while $k(x, x_k)$ is an RBF (Gaussian kernel) as specified in Equation (14).

$$k(x, x_k) = e^{-dk/2\sigma^2}, \, d_k = (x, x_k)^T (x, x_k) \tag{14}$$

where $d_k$ is the squared Euclidean distance between the input $x$ and training samples $x_k$.

*4.3. Analysis Steps of the Proposed ANN Algorithms*

Data need to be normalized to reduce errors. The data are collected and prepared for use in the model. This ANN model has four input nodes (User, PLR, Delay, and Spectral-Efficiency) and one output node (Fairness Factor). Then, the data are divided into two sections (i.e., training data and testing data). This study considered 50 data points for the model, where 80% of these points were used for training (i.e., 40 samples), while 20% were used for testing (i.e., 10 samples). Furthermore, the training (Data Train) and testing (Data Test) were separated, wherein Data Train = dataN for establishing the input and output data. The training NN model was divided into two steps, which are as follows: In Model 1; the NEWRB function is used for creating an RB network that approximates the function NEWRB for creating a two-layered network. Layer 1 is made up of a radial basis (RADBAS), neurons, and the weighted distribution (DIST) along with the net product (NETPROD) used to calculate the net input. Layer 2 is made up of pure line (PURELIN) neurons, whose weighted inputs are determined by DOT PRODuction (DOTPROD), while the net inputs are determined by NET SUM (NETSUM) [27]. The basis of the two layers is as follows:

"*P = P'; T = T';*"
"*eg = 0.02; sum-squared error goal*"
"*sc = 1; spread constant*"
"*net1 = NEWRB (P, T, eg, sc), Approximation NEWRBFNN*"

Model 2, i.e., the NEWRB function, generated a radial basis network that approximates the NEWRB function for generating a two-layered network. Layer 1 is made up of a radial basis function (RADBAS), neurons, and the distribution (DIST), which determines the weighted inputs, while the net production (NETPROD) calculates the net input. Layer 2

includes the pure line (PURELIN) neurons, which calculate the weighted input with the dot production (DOTPROD) and the net input with the net sum (NETSUM) [23]. The basis of the two layers is as follows:

*"P = P'; T = T";"*

*"eg = 0.02; sum-squared error target"*

*"sc = 1; spread constant"*

*"net2 = NEWRB (P, T, Spread), Exact NEWRBFNN"*

Model 3 uses the GRNN function approximation (NEWGRNN), which creates a two-layered network. Layer 1 comprises radial bases (RADBAS), neurons, and the distribution (DIST), which determines the weighted inputs, while the net production (NETPROD) calculates the net input. Layer 2 includes the pure line (PURELIN) neurons and determines the weighted input using the normalization product (NORMPROD), while the net input is determined using the net sum (NETSUM).

*"Spread = 0.7; "*

*"net3 = newgrnn (P, Spread);"*

*"A = sim (net3, P)."*

The testing model network replotted the training set and simulated the network responses for the input using the same range: (y-label, input vs. x-label, user) before an ANN model was proposed for the LTE downlink scheduling data (normalized data), wherein Figure 3 presents the output that has been tested and validated for deriving the results.

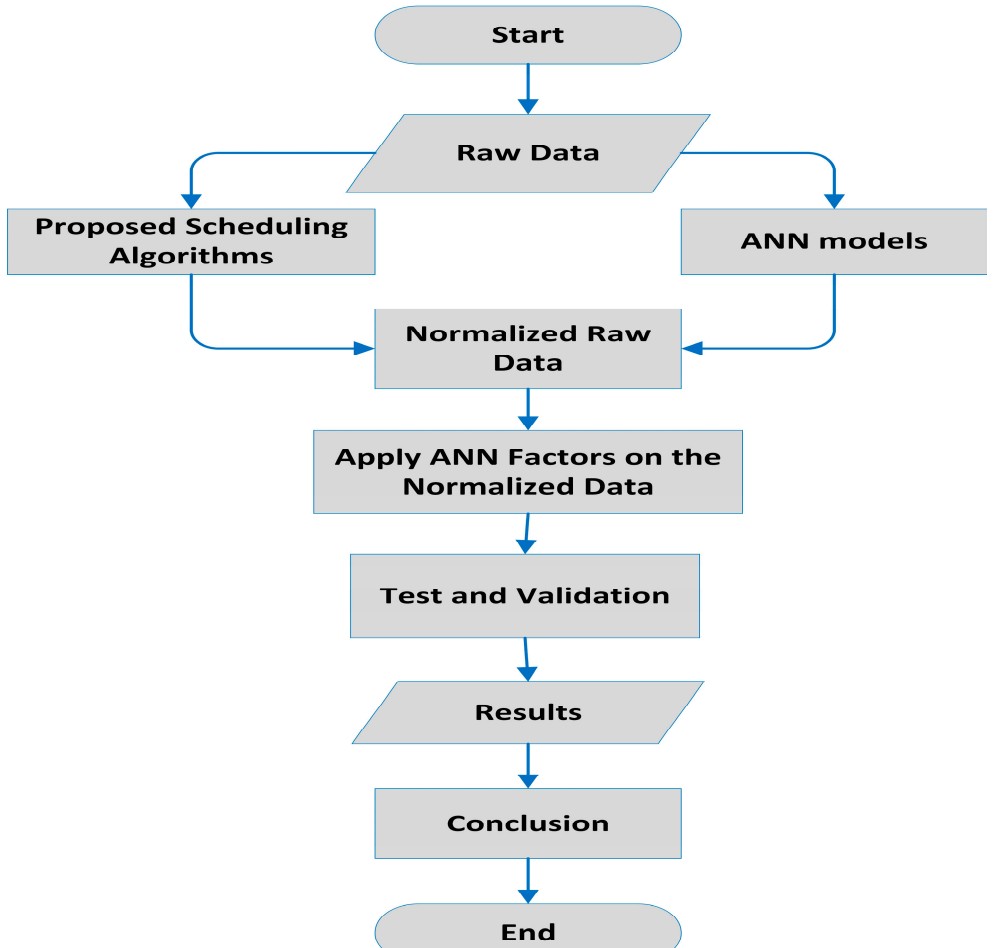

**Figure 3.** Flowchart of the Validation Process for the Normalized Data.

## 5. Experimental Results

A novel scheduling technique was implemented to address performance deterioration in the LTE network. This problem was addressed using the ANNA technique for radio resources. The researchers presented the results for the three LTE downlink scheduling processes' models. Detailed simulations were conducted with a network simulator, LTE Sim [24], that helped to assess the proposed algorithms' efficiency (PF, MLDWF, EXP-PF, and FLS). Table 1 presents the parameters used for the simulations. After considering seven cells with a fixed eNodeB, the simulator was run on an interference scenario for a single cell. Each cell had a radius of 1.5 km, and the number of users was randomly distributed in every cell, starting with three users. The interval between users was three, while the maximal number of users was 30.

**Table 1.** Simulation parameters used for the models.

| Parameters | Parameters Values | Parameters | Parameters Values |
|---|---|---|---|
| Number of clusters | 1 | Bandwidth | 10 MHz |
| Number of cells | 7 | simulation duration | 46 s |
| Frame Structure | FDD | Video bit-rate | 256 kbps |
| UEs number | 30 | Flow duration | 40 s |
| Cell Radius | 1.5 Km | Maximum delay | 0.1 |
| Speed of UE | | 3 km/h | |

The results will be analyzed and discussed based on implementing different LTE-DL scheduling algorithms, including PF, MLWDF, EXP-PF, and FLS. In addition, comparing three ANN algorithms titled RBFNN, Exact RBFNN, and GRNN, which were implemented and analyzed, the ANN-predicted LTE fairness index models were more accurate in the current research.

### 5.1. Normalized Proportional Fair (PF) Algorithm

- Approximate RBFNN Model-PF Algorithm

Figure 4 presents the results of comparing the normalized proportional fair (PF) algorithm with an approximate radial basis function neural network (RBFNN) model for downlink scheduling in an LTE network. The performance of the two algorithms was evaluated in terms of their precision in allocating radio resources to different users. The results presented in Figure 4 show that the RBFNN model was more precise than the PF scheduling algorithm, with an average precision of 0.086 (%). This means that the RBFNN model better allocated radio resources to different users more precisely and efficiently than the PF algorithm.

The results also showed that the performance of the PF algorithm varied depending on the number and heights of the users in the network. Specifically, the raw data for the PF algorithm showed that the number of users ranged from 3 to 27, with heights of 6, 12, and 30. The results suggest that the RBFNN model may be a more effective and precise algorithm for downlink scheduling in an LTE network compared to the traditional PF algorithm.

- Exact RBFNN Mode-PF Algorithm

Figure 5 presents the results of comparing the normalized proportional fair (PF) algorithm with an exact radial basis function neural network (RBFNN) model for downlink scheduling in an LTE network. The performance of the two algorithms was evaluated in terms of their accuracy in allocating radio resources to different users. According to the results presented in Figure 5, the exact RBFNN model was more accurate than the PF scheduling algorithm, with an accuracy of 0.090%. This means that the exact RBFNN model was better at allocating radio resources to different users more accurately and efficiently compared to the PF algorithm. The results also showed that the performance of the exact RBFNN model was symmetrical for all users, which means that it performed consistently

well for all users in the network. Overall, the results suggest that the exact RBFNN model may be a better algorithm for downlink scheduling in an LTE network than the traditional PF algorithm. However, as with the results presented in Figure 4, it is important to note that the results presented in Figure 5 may be specific to the particular network scenario and conditions used in the study and can be directly applicable to other network environments.

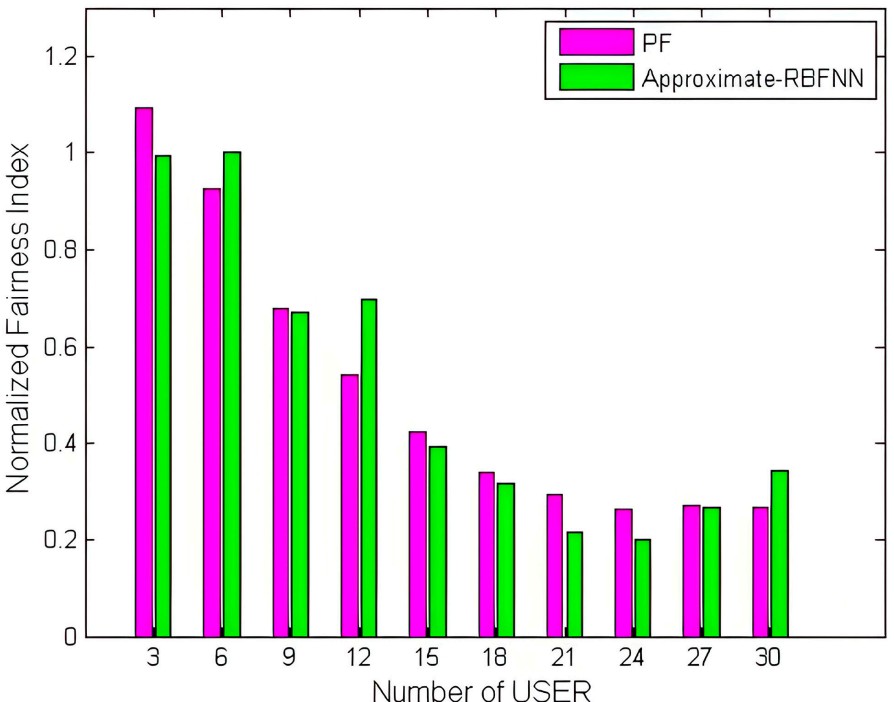

**Figure 4.** Comparison of performing the PF algorithm versus the approximate RBFNN model.

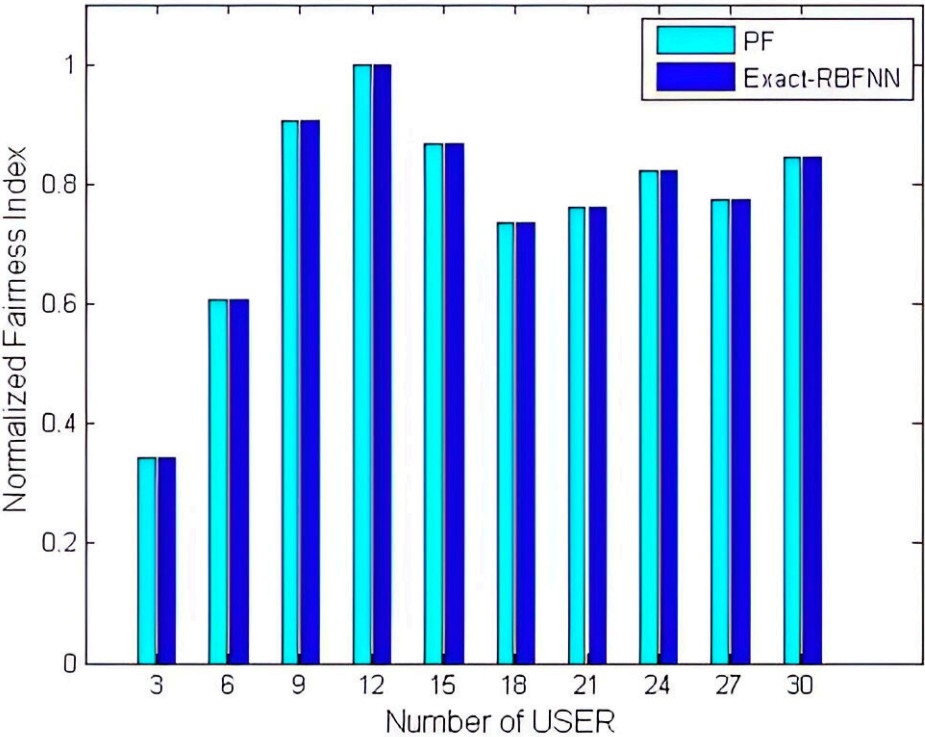

**Figure 5.** Comparison of performing the PF algorithm and exact RBFNN model.

- GRNN Model-PF Algorithm

Figure 6 presents the results of comparing the normalized proportional fair (PF) algorithm with a hybrid algorithm that combines the general regression neural network (GRNN) and radial basis function neural network (RBFNN) models for downlink scheduling in an LTE network. The performance of the two algorithms was evaluated in terms of their accuracy in allocating radio resources to different users. According to the results presented in Figure 6, the GRNN-RBFNN hybrid model was more accurate than the PF scheduling algorithm, with an accuracy improvement of 0.030%. This means that the GRNN-RBFNN hybrid model was better at allocating radio resources to different users more accurately and efficiently compared to the PF algorithm. The results also showed that the performance of the hybrid model was evaluated for different numbers of users, with the highest accuracy improvement observed for the numbers of users 9, 12, 15, 24, and 30. Overall, the results suggest that the GRNN-RBFNN hybrid model may be a more effective and accurate algorithm for downlink scheduling in an LTE network compared to the traditional PF algorithm. However, as with the results presented in Figures 4 and 5, it is important to note that the results presented in Figure 6 may be specific to the particular network scenario and conditions used in the study and can be directly applicable to other network environments.

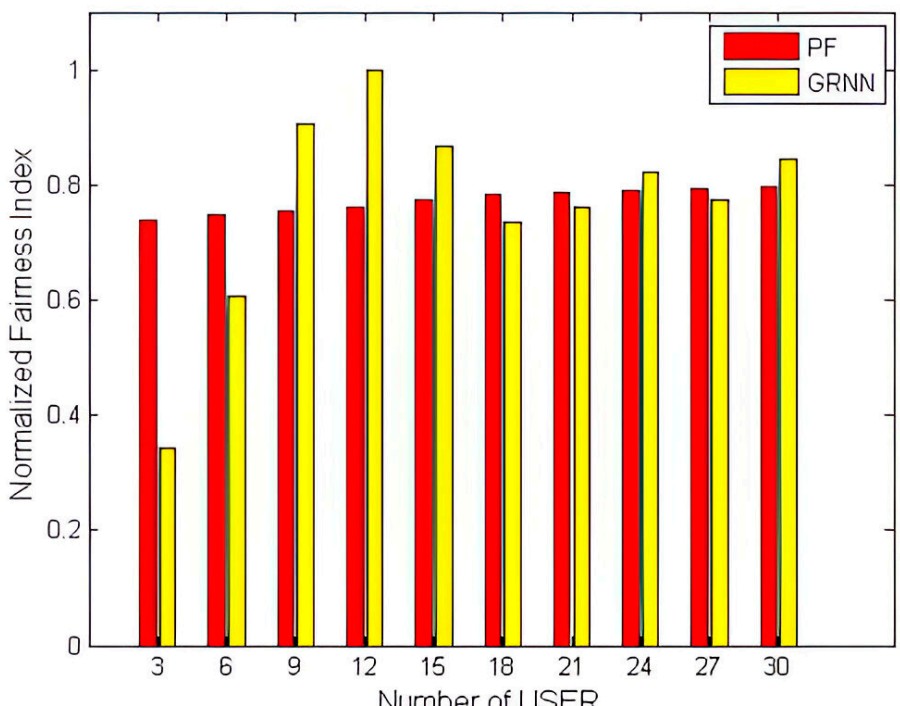

**Figure 6.** Comparison of the performance of the PF algorithm and GRNN model.

## 5.2. Normalized Maximum Largest Weighted Delay First (MLWDF) Algorithm

- Approximate RBFNN Mode-MLWDF Algorithm

Figure 7 presents the results of comparing the normalized maximum largest weighted delay first (MLWDF) algorithm by applying the approximate radial basis function neural network (RBFNN) model for downlink scheduling in an LTE network. The performance of the two algorithms was evaluated in terms of their accuracy in allocating radio resources to different users. According to the results presented in Figure 7, the approximate RBFNN model was more accurate than the MLWDF scheduling algorithm, with an accuracy improvement of 0.021%. This means that the approximate RBFNN model was better at allocating radio resources to different users more accurately and efficiently compared to the MLWDF algorithm. However, it is important to note that the accuracy improvement

of the RBFNN model was lower than the raw data presented by the MLWDF algorithm. The results also showed that the performance of the algorithms was evaluated for different numbers of users, with the highest value of the users indicated as 6, 18, and 30 compared to the raw data. Overall, the results suggest that the RBFNN model is a more effective and accurate algorithm for downlink scheduling in an LTE network compared to the MLWDF algorithm.

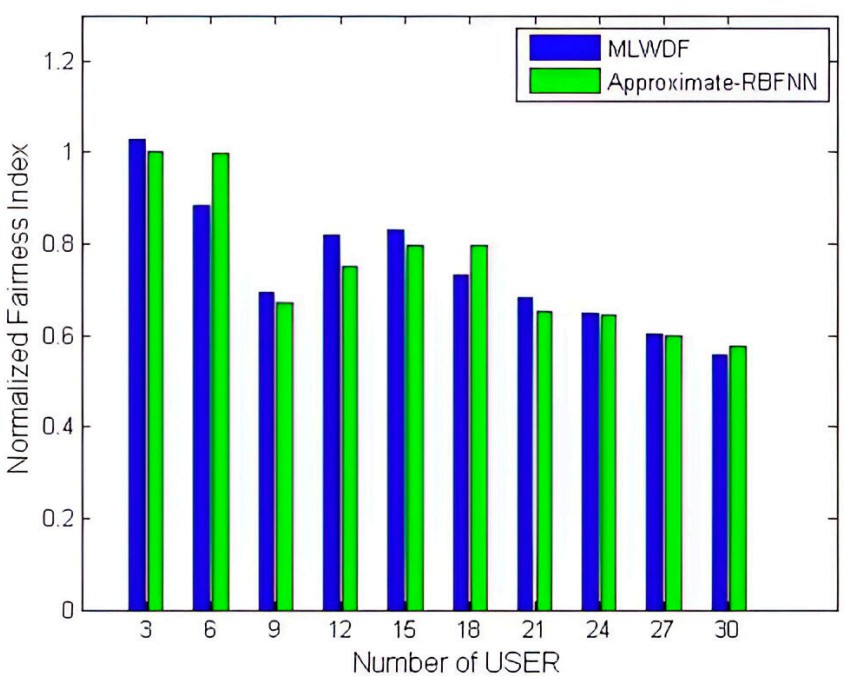

**Figure 7.** Comparison of performing the MLWDF algorithm and approximate RBFNN Model.

- Exact RBFNN Mode-MLWDF Algorithm

Figure 8 presents the results of comparing the normalized maximum largest weighted delay first (MLWDF) algorithm with an exact radial basis function neural network (RBFNN) model that was applied to the MLWDF for downlink scheduling in an LTE network. The performance of the two algorithms was evaluated in terms of their accuracy in allocating radio resources to different users. According to the results presented in Figure 8, the exact RBFNN model was more accurate than the MLWDF scheduling algorithm, with an average accuracy improvement of 0.021%. This means that the exact RBFNN model was better at allocating radio resources to different users more accurately and efficiently than the MLWDF algorithm. Additionally, the results indicate that the exact RBFNN model showed a symmetrical result compared to the raw data for all numbers of users tested. The results also showed that the performance of the algorithms was evaluated for different numbers of users tested in the study. Overall, the results suggest that the exact RBFNN model may be a more effective and accurate algorithm for downlink scheduling in an LTE network than the MLWDF algorithm.

- GRNN Model-MLWDF Algorithm

Figure 9 presents the results of comparing the normalized maximum largest weighted delay first (MLWDF) algorithm with a hybrid algorithm that uses both a generalized regression neural network (GRNN) and an exact radial basis function neural network (RBFNN) model for downlink scheduling in an LTE network. The performance of the two algorithms was evaluated in terms of their accuracy in allocating radio resources to different users.

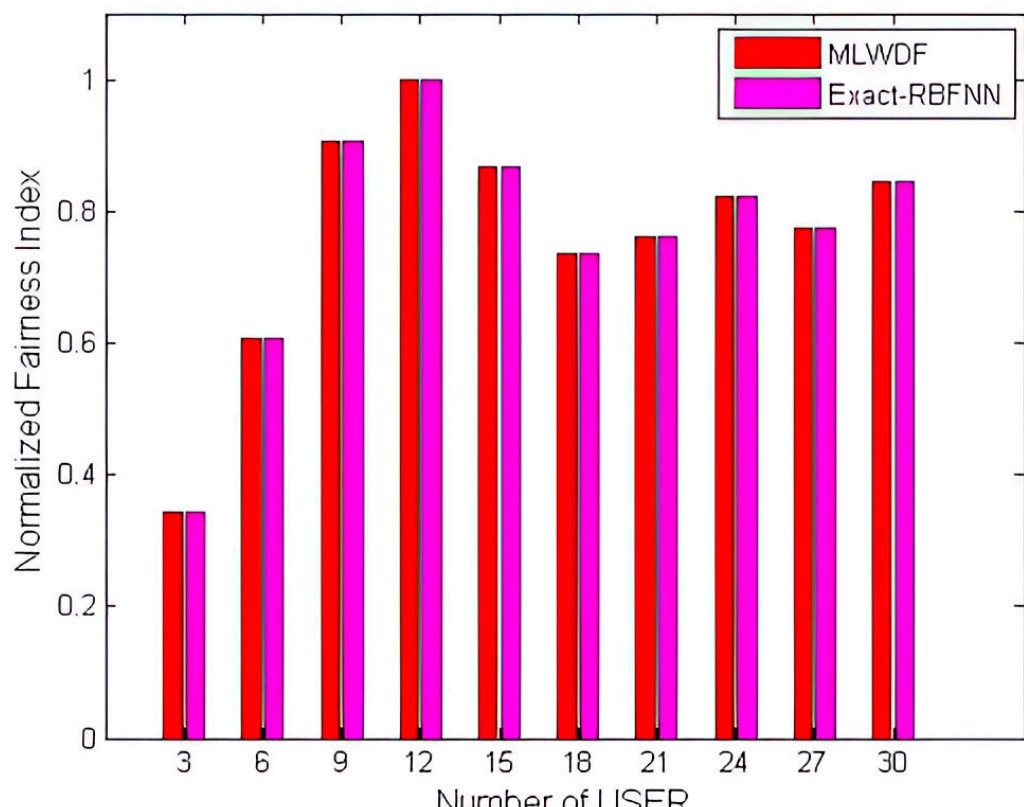

**Figure 8.** Comparison of performing the MLWDF algorithm and exact RBFNN model.

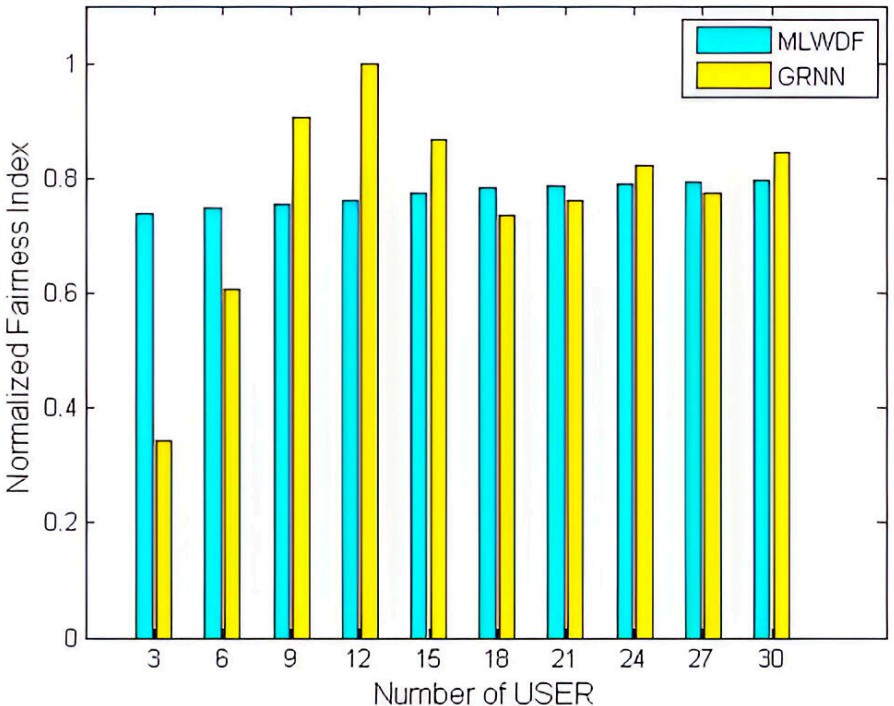

**Figure 9.** Comparison of performing the MLWDF algorithm and GRNN model.

According to the results presented in Figure 9, the hybrid GRNN-RBFNN model was much more accurate than the MLWDF scheduling algorithm, with an average improvement in accuracy of 0.030%. This means that the hybrid GRNN-RBFNN model was better at allocating radio resources to different users more accurately and efficiently compared to

the MLWDF algorithm. Additionally, the results indicate that the highest number of users tested was 30, with a symmetrical result for the raw data for all users tested except for 9, 12, 15, 24, and 30. Figure 9 may be specific to the particular network scenario and conditions used in the study and can be directly applicable to other network environments.

### 5.3. Normalized Exponential/Proportional Fairness (EXP-PF) Algorithm

- Approximate RBFNN Model-EXP-PF Algorithm

Figure 10 presents the results of comparing the normalized exponential/proportional fairness (EXP-PF) algorithm with an approximate radial basis function neural network (RBFNN) model for downlink scheduling in an LTE network. The performance of the two algorithms was evaluated in terms of their accuracy in allocating radio resources to different users.

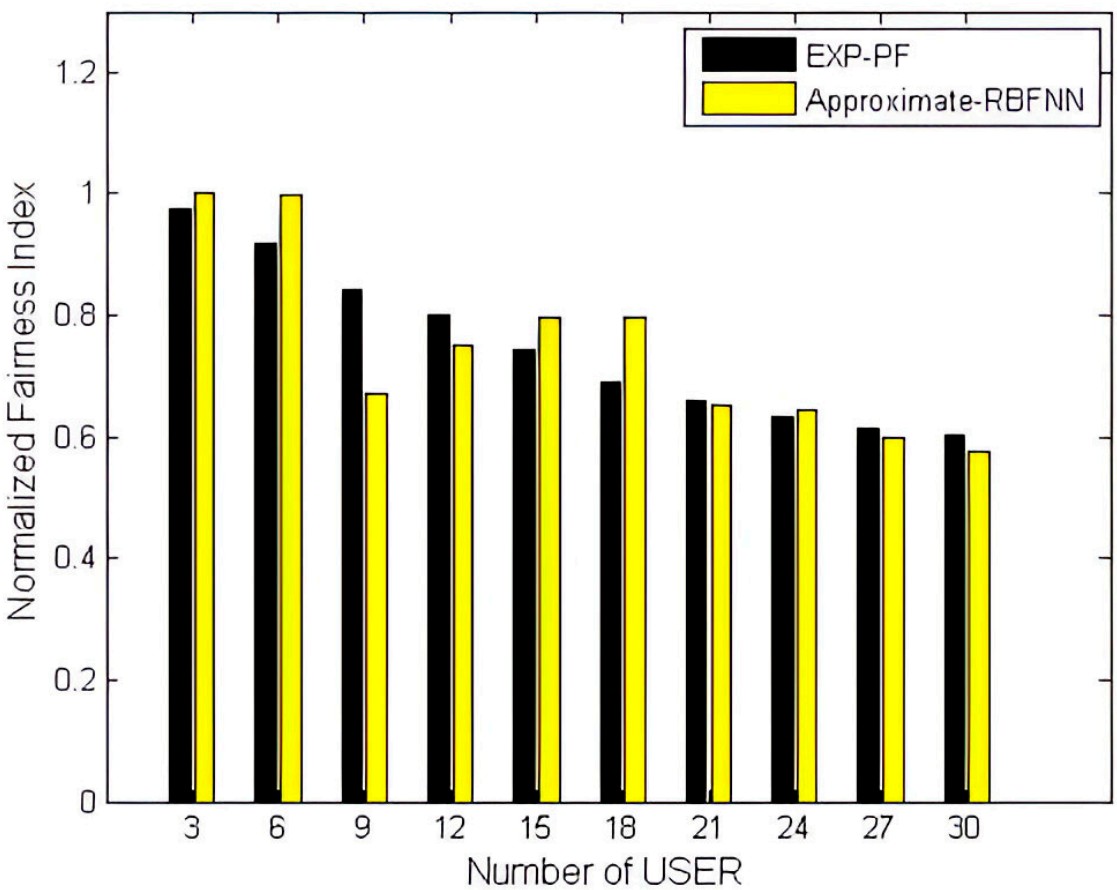

**Figure 10.** Comparison of performing the EXP-PF model and approximate RBFNN.

According to the results presented in Figure 10, the approximate RBFNN model was more accurate than the EXP-PF scheduling algorithm, with an improvement in accuracy of 0.021%. However, the approximate RBFNN model was less accurate than the raw data. The users tested included 9, 12, 21, 18, 27, and 30, with the highest value of users indicated as 3, 6, 15, 18, and 24 compared to the raw data. Figure 10 may be specific to the particular network scenario and conditions used in the study and can be directly applicable to other network environments.

- Exact RBFNN Model-EXP-PF Algorithm

Figure 11 presents the results of comparing the normalized exponential/proportional fairness (EXP-PF) algorithm with an exact radial basis function neural network (RBFNN) model for downlink scheduling in an LTE network. The performance of the two algorithms was evaluated in terms of their accuracy in allocating radio resources to different users.

According to the results presented in Figure 11, the exact RBFNN model was more accurate than the EXP-PF scheduling algorithm, with an improvement in accuracy of 0.023%. All users tested, including 3, 6, 9, 12, 15, 18, 21, 24, 27, and 30, presented symmetrical results compared to the raw data. It is important to note that the results presented in Figure 11 may be specific to the particular network scenario and conditions used in the study and can be directly applicable to other network environments.

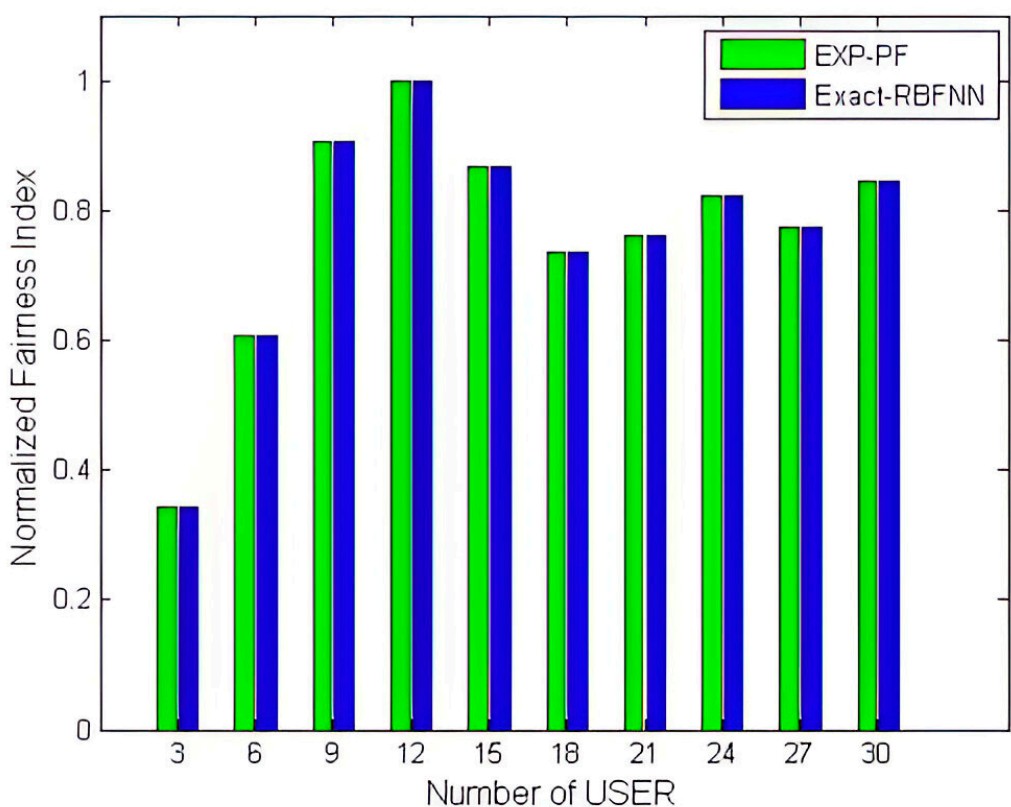

**Figure 11.** Comparison of performing the EXP-PF model and exact RBFNN.

- GRNN Model-EXP-PF Algorithm

Figure 12 presents the results of comparing the normalized exponential/proportional fairness (EXP-PF) algorithm with a hybrid model combining a generalized regression neural network (GRNN) and a radial basis function neural network (RBFNN) for downlink scheduling in an LTE network. The performance of the two algorithms was evaluated in terms of their accuracy in allocating radio resources to different users. According to the results presented in Figure 12, applying the proposed hybrid GRNN-RBFNN model for EXP-PF scheduling was more accurate than the EXP-PF scheduling algorithm, with an improvement in accuracy of 0.030%. The number of users tested was 3, 6, 18, 21, and 27, and the highest value of the users was indicated as 9, 12, 15, 24, and 30 compared to the raw data. It is important to note that the results presented in Figure 12 may be specific to the particular network scenario and conditions used in the study and can be directly applicable to other network environments. Further evaluation and validation of the hybrid GRNN-RBFNN model in a variety of network scenarios and conditions may be necessary to fully assess its effectiveness as a downlink scheduling algorithm in LTE networks.

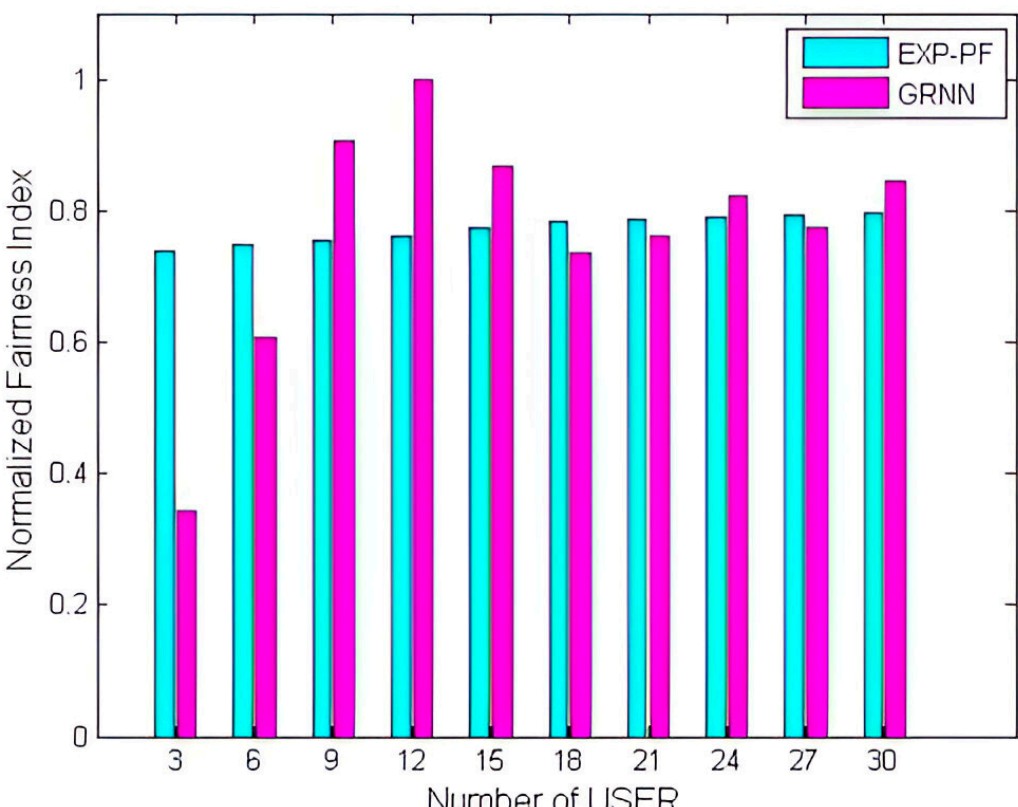

**Figure 12.** Comparison of performing the EXP-PF model and GRNN model vs. the number of users.

*5.4. Normalized Frame Level Scheduler (FLS) Algorithm*

- Approximate RBFNN Mode-FLS-Algorithm

The normalized frame level scheduler (FLS) algorithm is a scheduling algorithm used in LTE networks. Figure 13 shows the comparison between the proposed algorithm that applies an approximate RBFNN model on the LTE FLS algorithm and the FLS algorithm; the graph indicates that the approximate RBFNN model is more accurate than the FLS schedule method by approximately 0.11%. However, the approximate RBFNN model is still less accurate than the FLS's raw data. The users in this experiment were 9, 21, 24, 27, and 30. The highest value of the users was listed as 3, 6, 12, 15, and 18, which is different from the raw data.

- Exact RBFNN Model Mode-FLS-Algorithm

Figure 14 shows that the exact RBFNN (radial basis function neural network) model that was proposed for the FLS algorithm LTE network outperforms the FLS schedule algorithm by 0.011% in terms of accuracy. This is because the exact RBFNN model was able to predict optimal scheduling policies for all users in the experiment, and the results were the same when compared to the raw data in the FLS. It is important to note that the accuracy difference between the two algorithms may seem small (0.011%), but in practical applications, even small improvements in accuracy can lead to significant improvements in network performance and user experience.

- GRNN Mode-FLS-Algorithm

Figure 15 shows that the GRNN-RBFNN (general regression neural network radial basis function neural network) model that was implemented for the FLS algorithm is more accurate than the FLS schedule algorithm by 0.030%. This indicates that the GRNN-RBFNN model is better than the FLS algorithm at predicting optimal scheduling policies for different users. It is also noted that the numbers of users involved in the experiment were 3, 6, 18, and 21, and their corresponding highest values were listed as 9, 12, 15, 24,

and 30, respectively. The results indicate that the fairness index for some users was lower compared to others. This suggests that the scheduling algorithm used may not be able to provide equal QoS to all users, and further optimization may be required to improve fairness. It is important to note that the results presented in Figure 15 are specific to the experiment or study being referred to and can be applied to other scenarios or settings.

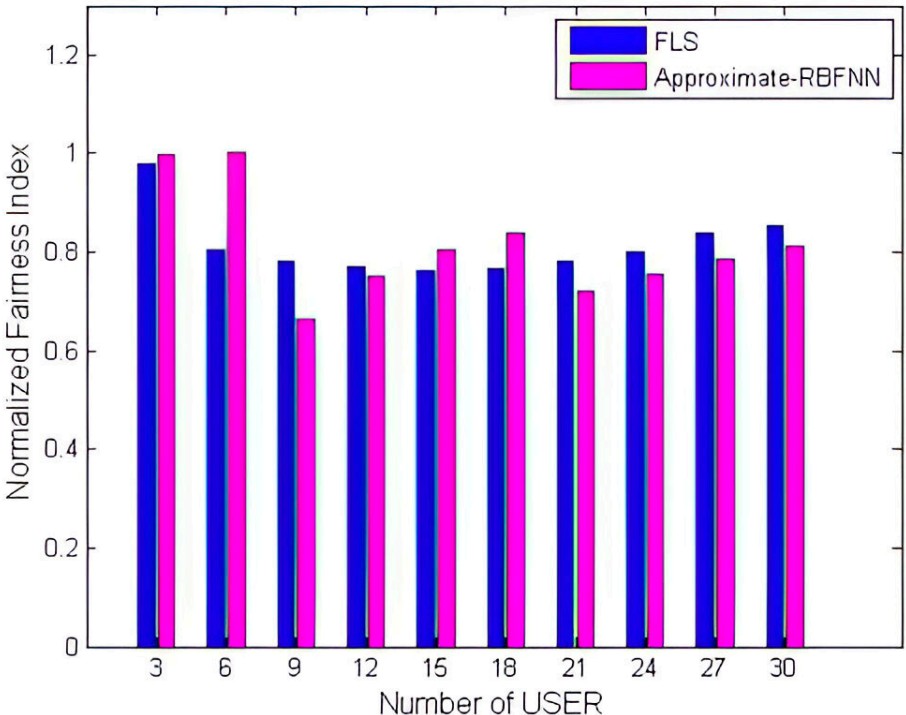

**Figure 13.** Comparison of performing the FLS model and approximate RBFNN model.

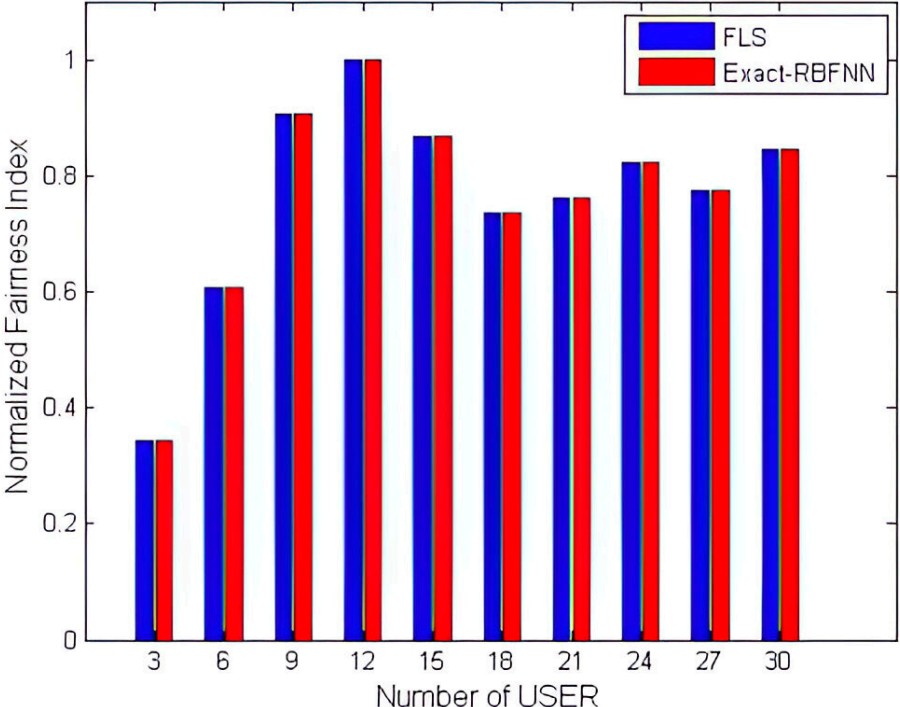

**Figure 14.** Comparison of performing the FLS model and exact RBFNN model.

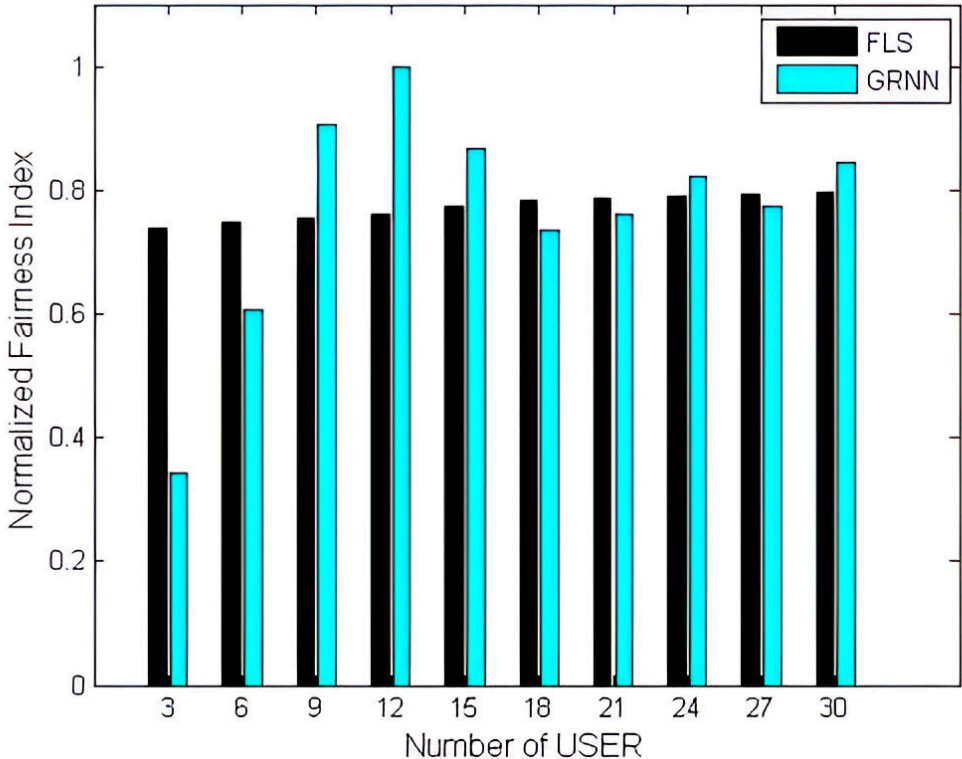

**Figure 15.** Comparison of performing the FLS model and GRNN -RBFNN model.

## 6. Conclusions

The issues related to the allocation of resources over LTE networks are regarded as a major challenge that can provide satisfactory QoS to the active users for every transmission time interval (TTI). The resources that are available on the network are distributed amongst all the users in such a way that they can fulfil their needs. The researchers investigated the different scheduling algorithms (such as PF, MLDWF, EXP-PF, and FLS) based on the LTE fairness index. For this purpose, they proposed a novel scheduling algorithm for improving the low-performance deterioration of the network by applying the ANN algorithm technique, i.e., the approximate radial basis function neural network (RBFNN), the exact RBFNN, and the generalized regression neural network (GRNN). The exact Gaussian fields are approximated using the appropriate radial function (RF) for different radio services. This study investigated the performance of a popular resource allocation algorithm used in the LTE downlink stream to satisfy the QoS requirements. The results indicated that the proposed method improved performance over the other algorithms based on all calculated metrics. The algorithm indicated an improved percentage value and a higher fairness factor. Moreover, to overcome fundamental challenges like high data rate real-time traffic, high QoS, reliable and robust communication links, and an ever-increasing demand for network services, future wireless networks will depend on the new technological advancement known as ANNs. Today's mobile or cellular consumers are increasingly utilizing multimedia services. With the increased data rate based on ANNs and machine learning, seamless multimedia services are now accessible.

**Author Contributions:** Conceptualization, A.A.M., F.Y.H.A. and B.K.; methodology, A.A.M., F.Y.H.A., B.K., D.A.Z. and J.H.Y.; software, A.A.M. and F.Y.H.A.; formal analysis, A.A.M., F.Y.H.A., B.K., D.A.Z. and J.H.Y.; investigation, A.A.M. and F.Y.H.A.; resources, D.A.Z. and J.H.Y.; writing—original draft preparation, A.A.M., F.Y.H.A., B.K., D.A.Z. and J.H.Y.; writing—review and editing, A.A.M., F.Y.H.A., B.K., D.A.Z. and J.H.Y.; project administration, A.A.M. and F.Y.H.A. All authors have read and agreed to the published version of the manuscript.

**Funding:** The research leading to these results has received a research grant from the Ministry of Higher Education, Research and Innovation (MoHERI) of the Sultanate of Oman under the Block Funding Program. MoHERI Block Funding Agreement No: MoHERI/BFP/SU/01/2021.

**Data Availability Statement:** Data is available on request due to restrictions on privacy.

**Acknowledgments:** The corresponding author would like to thank all other contributors for their guidance in an advisory position and for helping to develop an attitude helpful to scholarly research and development. The authors do not have competing interests.

**Conflicts of Interest:** All authors have stated that they have no competing interests. The sponsors had no input into the study's conception or execution, data acquisition, analysis, interpretation, article preparation, or final publication decision.

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
