# Peer review of "Optimized Downlink Scheduling over LTE Network Based on Artificial Neural Network"

_computers, doi:10.3390/computers12090179_

Round 1
Reviewer 1 Report
-
Introduction: The introduction of the paper lacks a clear problem statement and research gap. The authors should explicitly state the specific problem their research is addressing and identify the gap in the existing literature that their study aims to fill. This will provide a clear context for the study and help readers understand the significance of the research.
-
Literature Review: The literature review lacks a clear structure and critical analysis. The authors should organize the literature review around key themes or topics related to their research and provide a critical analysis of the previous studies. This will help to establish the context for the study and demonstrate its contribution to the field.
-
Validation and Comparison with Existing Models: The paper lacks a comprehensive validation of the proposed model. The authors should provide a detailed comparison of their proposed model with existing models in terms of performance metrics. This will help to establish the effectiveness of the proposed model. Additionally, the authors should use a separate validation dataset to ensure the model's generalizability.
-
Selection of Normalization Techniques: The authors have not provided a clear rationale for the selection of the normalization techniques used in the study. They should provide a detailed explanation of why these techniques are suitable for their problem and discuss their potential impact on the performance of the ANN model.
-
Statistical Analysis: The paper lacks statistical analysis to support the claims made. The authors should conduct statistical tests to confirm the significance of the improvements in accuracy or precision between the algorithms. This will strengthen the validity of the results and provide a more robust comparison of the algorithms.
-
Generalizability of the Results: The results of the study may not be directly applicable to other network environments, which limits their generalizability. The authors should conduct additional simulations under different network conditions and scenarios to demonstrate the robustness and general applicability of their findings.
-
Discussion on Limitations and Future Work: The conclusion section lacks a detailed discussion on the limitations of the study and potential areas for future work. The authors should acknowledge the limitations of their proposed method and discuss potential areas for future research. This will provide a more balanced view of the research and help readers understand the context in which the results should be interpreted.
The quality of the English language in the manuscript is generally good, with clear and concise sentences. However, there are a few areas where the language could be improved for clarity and readability. For instance, some sentences are overly long and complex, which could potentially confuse readers. It would be beneficial to break these down into shorter, more digestible sentences. Additionally, there are a few instances of awkward phrasing and minor grammatical errors that should be corrected.
Author Response
Thank you for the effort and valuable comments.
Please see the attachment.

Reviewer 2 Report
computers-2551157
This article focuses on the LTE downlink scheduling algorithms to propose a new technique that derives precise and trustworthy data output to meet the requirements of the 3GPP-proposed LTE network standard. The suggested technique has been developed using a combination of two artificial neural network (ANN) algorithms: the normalized radial basis function neural network (RBFNN) and the generalized regression neural network (GRNN) algorithms. Based on the new technique's findings, the proposed algorithm; in addition to having the potential to considerably enhance the efficiency of real-time streaming over the common LTE downlink algorithms, also has a reduced computational complexity. This paper compares the proposed technique to LTE downlink algorithms regarding the packet loss rate, delay, spectrum efficiency, and fairness. Some of my comments are as follows:
1. Section 1 “introduction” section is very lengthy and wordy. Also, section “4. Proposed Model” should be split into 2-3 brief and concise paragraphs. The current one seems too wordy and confusing. Similarly, revise the remaining sections and avoid wordy and confusing paragraphs.
2. There are numerous grammatical mistakes and typos that must be corrected with detailed proofreading.
3. The experimental simulation results of this work are very limited
4. Most of the figures seems snapshots, the authors should add original figures. The current figures resolution is very low.
5. In the term “Where” after equation (2), (3), and so on, “w” should be in small letters as “where”
6. There are several numerical equations, but most of them are missing proper explanation and citations.
7. The novelty of this work is very limited, the authors have just varied some parameters and added a separate figure for it.
8. The technical depth of the paper is not adequate too.
9. The references seem limited and most of them are old. It is better to add more references, especially from the last 3-5 years.
Extensive editing of English language required
Author Response

(The authors gave the same response as above.)

Reviewer 3 Report
The manuscript presents a novel LSTM downlink scheduling algorithm based on ANN, RBFNN, and GRNN models. However, there are several major issues that need to be addressed as follows:
1) Lack of comparison with prior related works to establish the novelty and advantages of the proposed method.
2) Insufficient discussion of the pros and cons of the proposed method, which would provide better insights into its effectiveness and limitations.
3) The conclusion needs to be more comprehensive and conclusive, leaving no ambiguity regarding the findings.
4) The manuscript contains numerous typos and grammar mistakes, and it is essential to thoroughly proofread it before resubmitting.
5) Additionally, in the abstract, the abbreviation "PF" is used, but its meaning is not clarified. Please provide stand for it.
6) The usage of ";" appears to be arbitrary in many sentences
7) The references are not correct MDPI format yet.
Extensive editing of English language required
Author Response

(The authors gave the same response as above.)

Round 2
Reviewer 1 Report
The authors have successfully answered to all my questions.
Author Response
I want to take this opportunity to thank the editorial office members and the reviewers for their valued comments and feedback, which enriched the article.
Reviewer 2 Report
Most of my previous comments still apply to the revised version because there is no such modifications according to the previous comments, especially at novelty and experimental results. Moreover, the author's haven't revised the article carefully, section 2 and so on still exists those lengthy, wordy and confusing large large paragraphs which makes the reader confused.
Moderate editing of English language required
Author Response
I want to take this opportunity to thank the editorial office members and the reviewers for their valued comments and feedback, which enriched the article.
Kindly refer to the response provided in the attached document.

Reviewer 3 Report
Authors addressed my concerns. Accept for publication.
Author Response

(The authors gave the same response as above.)

Round 3
Reviewer 2 Report
NA